# Colorectal Cancer Screening Outcomes of 2412 Prostate Cancer Patients Considered for Carbon Ion Radiotherapy

**DOI:** 10.3390/cancers13174481

**Published:** 2021-09-06

**Authors:** Nao Kobayashi, Takahiro Oike, Nobuteru Kubo, Yuhei Miyasaka, Tatsuji Mizukami, Hiro Sato, Akiko Adachi, Hiroyuki Katoh, Hidemasa Kawamura, Tatsuya Ohno

**Affiliations:** 1Department of Radiation Oncology, Gunma University Graduate School of Medicine, 3-39-22 Showamachi, Maebashi 371-8511, Japan; m10201207@gunma-u.ac.jp (N.K.); kubo@gunma-u.ac.jp (N.K.); akiadachi@gunma-u.ac.jp (A.A.); tohno@gunma-u.ac.jp (T.O.); 2Gunma University Heavy Ion Medical Center, 3-39-22 Showa-machi, Maebashi 371-8511, Japan; y.miyasaka@gunma-u.ac.jp (Y.M.); hiro.sato@gunma-u.ac.jp (H.S.); kawa@gunma-u.ac.jp (H.K.); 3Department of Radiology, University of Toyama, 2630 Sugitani, Toyama 930-0194, Japan; tmizukam@med.u-toyama.ac.jp; 4Department of Radiation Oncology, Kanagawa Cancer Center, 2-3-2 Nakao, Asahi-ku, Yokohama 241-8515, Japan; hkatoh@kcch.jp

**Keywords:** colorectal cancer, screening, prevalence, prostate cancer, radiotherapy

## Abstract

**Simple Summary:**

Colorectal cancer (CRC) screening is effective for cancer detection in average-risk adults. For prostate cancer (PCa) patients considered for carbon ion radiotherapy (CIRT), pre-treatment CRC screening is performed empirically to avoid post-treatment colonoscopic manipulation. However, the outcomes of screening remain unclear. To address this, we analyzed the outcomes of 2412 PCa patients at average risk for CRC who underwent routine pre-CIRT CRC screening and found that the estimated CRC prevalence was greater than that reported by 17 previous large-scale screening studies analyzing average-risk adults. These data indicate the possibility that the prevalence of CRC in PCa patients is greater than that in general average-risk adults, warranting further research.

**Abstract:**

Colorectal cancer (CRC) screening is effective for detecting cancer in average-risk adults. For prostate cancer (PCa) patients considered for carbon ion radiotherapy (CIRT), pre-treatment CRC screening is performed empirically to avoid post-treatment colonoscopic manipulation. However, the outcomes of screening this population remain unclear. Here, we compared the outcomes of routine pre-CIRT CRC screening of 2412 PCa patients at average risk for CRC with data from two published datasets: the Japan National Cancer Registry (JNCR) and a series of 17 large-scale screening studies analyzing average-risk adults. The estimated prevalence rate was calculated using the pooled sensitivity elucidated by a previous meta-analysis. Consequently, 28 patients (1.16%) were diagnosed with CRC. CRC morbidity was significantly associated with high pre-treatment levels of prostate-specific antigen (*p* = 0.023). The screening positivity rate in this study cohort exceeded the annual incidence reported in the JNCR for most age brackets. Furthermore, the estimated prevalence rate in this study cohort (1.46%) exceeded that reported in all 17 large-scale studies, making the result an outlier (*p* = 0.005). These data indicate the possibility that the prevalence of CRC in PCa patients is greater than that in general average-risk adults, warranting further research in a prospective setting.

## 1. Introduction

Colorectal cancer (CRC) and prostate cancer (PCa) are the third and fifth leading causes of death, respectively, among all cancers in men worldwide [1]. For localized PCa, radiation therapy is the standard treatment [2]; however, carbon ion radiotherapy (CIRT) shows promise as a definitive treatment that achieves a 5-year cause-specific survival of 99% [3]. Radiation-induced proctitis is an adverse effect that develops in patients with PCa treated with CIRT [4], and subsequent colonoscopic manipulation of the irradiated site can exacerbate this rectal condition [5]. However, CRC screening using the fecal immunochemical test (FIT), followed by colonoscopy for FIT-positive cases, is recommended for average-risk adults [6,7]. Therefore, it is preferrable that patients with PCa who are considered for CIRT receive CRC screening prior to treatment. From this perspective, CRC screening is routine practice for PCa patients on their first referral to our CIRT center. However, the incidence and clinical course of CRC, and its associated clinical factors, in this population remain unclear. To address this, we analyzed outcomes after routine pre-treatment CRC screening performed at our center.

## 2. Materials and Methods

### 2.1. Study Design

Patients with pathologically-confirmed PCa who were at average risk for CRC [7] and referred to Gunma University Heavy Ion Medical Center (GHMC, Maebashi, Japan) in 2012–2020 were enrolled retrospectively. Medical charts were reviewed retrospectively and the outcomes of CRC screening were recorded. The association between CRC morbidity and clinical risk factors for PCa (i.e., initial prostate-specific antigen (PSA) value, T stage (based on the TNM classification of the International Union Against Cancer 2009), Gleason score, and tumor risk group) was analyzed [8,9].

The outcomes of CRC screening in this study cohort were compared with those in two datasets. One was the Japan National Cancer Registry 2015 (JNCR) [10]. The strength of this dataset is that it reports the real-world incidence of CRC and enables a nation-, gender-, and age-matched comparison with the present study cohort. The other is a series of 17 large-scale studies presented in the consensus statement for CRC screening published by the U.S. Multi-Society Task Force (USMSTF) (Table 1) [6,11,12,13,14,15,16,17,18,19,20,21,22,23,24,25,26,27]. These studies report the outcomes of CRC screening in an average-risk population comprising more than 1000 participants (median, 4202; range, 1204–27,860). The strength of this dataset is the robustness of the reported prevalence rate, i.e., in all studies, the prevalence of CRC was confirmed either by colonoscopy or by 2-year follow up.

The estimated prevalence of CRC in the present study cohort was calculated by dividing the raw screening positivity by 0.79; this number was based on the results of a meta-analysis of 19 studies of average-risk adults that reported the pooled sensitivity of CRC screening in average-risk adults as 0.79 (95% confidence interval, 0.69–0.86) [28].

### 2.2. Colorectal Cancer Screening

All participants received CRC screening at the first referral to GHMC. This screening comprised a two-sample FIT followed by colonoscopy for cases with at least one FIT-positive result [6,7]. If a patient had received screening within 1 year of the first referral to GHMC, the results were used without performing a new test; therefore, CRC screening in a subset of participants was carried out at institutes other than Gunma University. The definition of advanced adenoma, as well as the indications for resection of polyps, was not standardized among institutes. For a maximally precise interpretation of the report contents from this perspective, the colonoscopy findings were recorded using the following classification: CRC (i.e., invasive carcinoma), resected adenoma, unresected polyp, and other benign findings.

### 2.3. Statistical Analysis Subsection

Outliers were tested using the Smirnov–Grubbs test after confirming the normality of the dataset using the Shapiro–Wilk test. The association between CRC morbidity and clinical risk factors for PCa was examined using Fisher’s exact test. The level of statistical significance was set at *p* < 0.05 after Bonferroni correction. Statistical analysis was performed using R (R Foundation for Statistical Computing, Vienna, Austria) on the EZR platform [29].

## 3. Results

Of the 2555 consecutive PCa patients enrolled in the study, 2537 were at average risk for CRC; 125 patients were excluded due to incomplete medical records (including the absence of screening information). Therefore, 2412 patients were analyzed (Figure 1). The average age was 69 years (range, 50–94 years). Initial PSA values, T stage, Gleason score, and PCa risk group are summarized in Table 2. Of all patients, 604 patients (25.0%) were positive for FIT; of these, 586 (97.0%) underwent colonoscopy. Figure 2 shows the colonoscopy findings of FIT-positive patients. CRC was detected in 28 (1.16%) patients. Meanwhile, non-malignant findings associated with lower gastrointestinal hemorrhage were identified in 421 (17.4%) patients.

All 28 patients diagnosed with CRC prioritized treatment for CRC before treatment for PCa; 24 patients (85.7%) completed CIRT successfully after treatment for CRC. Of the remaining four patients, two received chemotherapy; therefore, CIRT for PCa was not indicated. Of the remaining two patients, one had pelvic lymph node involvement and was not considered eligible for CIRT, and the other received surgery for CRC and at the time of writing is considering CIRT for PCa when his health recovers. Of note, none of the 24 patients treated with CIRT for PCa after treatment for CRC developed symptomatic proctitis post-CIRT.

The association between CRC morbidity and major clinical risk factors for PCa was analyzed. A high initial PSA value showed a significant association with CRC morbidity (*p* = 0.023), whereas T stage, Gleason score, or tumor risk group did not (Table 2). Age was not associated with CRC morbidity (69.6 ± 6.4 vs. 69.2 ± 6.8 for CRC patients and non-CRC patients, respectively; *p* = 0.62).

The outcomes of CRC screening in this study cohort were compared with the nation-matched JNCR dataset (see Section 2.1 for details). In the JNCR dataset, there was a trend toward a higher incidence of CRC among the elderly (Figure 3a). The same trend in the screening positivity rate was observed in this study cohort, except for age brackets 50–54 years and ≥80 years; it is likely that these exceptions are due to the small number of analyses. The screening positivity rate in this study cohort exceeded the annual incidence of CRC for all age brackets observed in the JNCR dataset; the exception was the ≥80 years population (Figure 3a). These data indicate that the performance of CRC screening in this study cohort is robust.

The estimated prevalence rate of CRC in this study cohort was 1.46%, based on the pooled sensitivity for CRC screening among asymptomatic average-risk adults (i.e., 0.79; see Section 2.1 for details). Interestingly, the estimated prevalence rate of CRC in this study cohort was greater than that in any of the 17 large-scale cohorts of average-risk adults, making the current cohort an outlier (*p* = 0.005, see Section 2.1 for details of the control cohorts) (Figure 3b). These data indicate that the prevalence of CRC in PCa patients considered for CIRT is greater than that in average-risk adults.

## 4. Discussion

Here, we found that the prevalence of CRC in PCa patients referred to GHMC for the purpose of CIRT was greater than that in the general population. To the best of our knowledge, this is the first study to report a greater prevalence rate of CRC in PCa patients compared with a risk-matched general population. The comparison with the USMSTF-reported studies was performed under risk-matched settings. Meanwhile, the comparison with the JNCR dataset was performed under nation-, gender-, and age-matched settings. These data indicate the robustness of the main finding of this study.

A possible explanation for the high prevalence of CRC in our study is that CRC and PCa share the same potential risk factors. Risk factors can be classified as endogenous or exogenous, although some factors are not exclusively one or the other (e.g., race, aging, and oxidative stress). Current alcohol intake was associated with an increased risk of CRC if one consumed an average of one or more alcoholic drinks per day [30], and the risk increased with alcohol intake for PCa as well [31]. It has been suggested that a diet high in meat-derived fats may also increase the risk of advanced colorectal neoplasia [30]. Fat intake, especially polyunsaturated fats, has also been reported to be strongly positively correlated with incidence and mortality in PCa [31], and this mechanism may be due to fat-induced changes in hormone profiles [32], the effects of fat metabolites as protein and DNA reactive intermediates [33], or elevated oxidative stress due to fat [34]. It has been reported that dietary fiber intake from cereals was associated with a lower relative risk of advanced neoplasia of the colon [30]. Cereal consumption has also been shown to be inversely associated with mortality in PCa [31]. It is suggested that plant ligands and isoflavonoids in grains and cereals are converted by intestinal bacteria into compounds with weak estrogenic and antioxidant activity, which have anticancer effects [35]. With regard to vitamin D intake, it has also been suggested to be inversely related to the risk of CRC [30]. In PCa, 1α-25-dihydroxyvitamin D (1,25-D), the hormonal form of vitamin D, has been reported to inhibit PCa cell invasion in vitro [36] and to exhibit antiproliferative and differentiation-promoting effects in the Dunning rat PCa model [37], suggesting that vitamin D deficiency has been a potential risk factor for PCa [31]. It is not likely that a potential age bias in this study cohort affected the greater CRC morbidity, considering that the trend of CRC morbidity by age group in this study cohort was consistent with that in the JNCR. It may be possible that this study cohort suffered from selection bias in that it included more health-conscious patients that favored CIRT, which is a promising, albeit scarce medical resource; even if that was the case, the bias should not have caused an increase (rather, it would have caused a decrease) in the prevalence of CRC associated with more frequent routine cancer screening of such patients.

Another possibility is the presence of unknown host genetic variants that contribute to predisposition to the two cancers; indeed, a subset of genes is responsible for cancer predisposing syndromes (e.g., the BRCA genes responsible for Hereditary Breast and Ovarian Cancers) [38]. In fact, tyrosine kinase receptors, which was isolated as an oncogene in CRC, is also expressed in PCa [39]. Tyrosine kinase receptors are thought to transmit proximal signals for neurotrophin-mediated growth in cancer, and the tyrosine kinase inhibitor K 252a inhibited the growth of cancer cell lines in vitro [40], further supporting the role of tyrosine kinase receptors in neurotrophin-mediated growth of PCa. Further research is needed to explore factors predictive of PCa sub-populations at high risk of CRC.

In this study, a high initial PSA value was associated with high CRC-related morbidity. Additionally, most immunohistochemical studies have concluded that abnormalities in the tumor suppressor gene *TP53*, which functions as a regulator of the cell cycle and is associated with many human malignancies, including colorectal cancer, are usually present in localized prostate cancers with high Gleason scores (>7) [41,42,43]. From this perspective, simultaneous screening for such common cancers is of high importance [44,45]. More importantly, there are no established treatment strategies for synchronous CRC and PCa. Corbin et al. performed a retrospective review of medical records at Duke University Medical Center and the Durham Veterans Affairs Medical Center between 1988 and 2017, and identified 54 patients with synchronous rectosigmoid cancer and PCa [46]. Seretis et al. performed a literature review of synchronous rectal cancer and PCa, and summarized 23 cases [47]. These studies report various (i.e., unstandardized) treatments for each patient. In this study, we report a series of 24 patients with synchronous CRC (including 14 patients of rectosigmoid cancer) and PCa who were treated successfully with surgical resection for the former, followed by CIRT for the latter. Furthermore, CIRT resulted in no symptomatic proctitis.

The screening positivity for CRC in our study cohort was markedly higher than the annual incidence of CRC in the JNCR dataset (1.16% vs. 0.37%, respectively) [10]. It is easy to imagine that in a real-world setting a portion of cases diagnosed with CRC would have been dropped from registration in the JNCR; therefore, it would be crude to compare the screening outcomes of this study cohort with the incidence reported in JNCR. Nevertheless, the incidence of CRC in the JNCR dataset is broadly consistent with the prevalence of CRC observed in nation-matched large-scale screening studies (i.e., median, 0.36%; range, 0.32–0.61%) [11,15,17,21,25], providing a certain level of justification for the current comparison. Radiotherapy for localized PCa may be postponed by performing androgen deprivation therapy [48]; therefore, detection of CRC prior to radiotherapy for PCa is of benefit to patients. In addition, in this study, we found benign lesions associated with lower gastrointestinal hemorrhage in 17.4% of participants; such information will be useful during post-CIRT follow up to avoid post-treatment colonoscopic manipulation that can exacerbate radiation-induced proctitis. Taken together, the results of the present study suggest that pre-treatment CRC screening is beneficial for PCa patients considered for CIRT. This benefit may be interpreted broadly with respect to other external beam radiotherapies based on photons or protons because, similar to CIRT, these modalities can cause radiation-induced proctitis in patients with localized PCa.

The study has several limitations. First, we were unable to standardize the details of CRC screening in terms of the brand and cut-off value of FIT, or the method and reporting of colonoscopy; this is because screening was performed at multiple institutes. Additionally, the clinical and epidemiological information is limited to PCa progression and to standard risk groups for CRC.

## 5. Conclusions

To elucidate the outcomes of CRC screening in PCa patients considered for CIRT, we performed a large-scale retrospective analysis of 2412 participants at average risk for CRC. These data indicate the possibility that the prevalence of CRC in PCa patients is greater than that in general average-risk adults, warranting further research in a prospective setting.

## Figures and Tables

**Figure 1 cancers-13-04481-f001:**
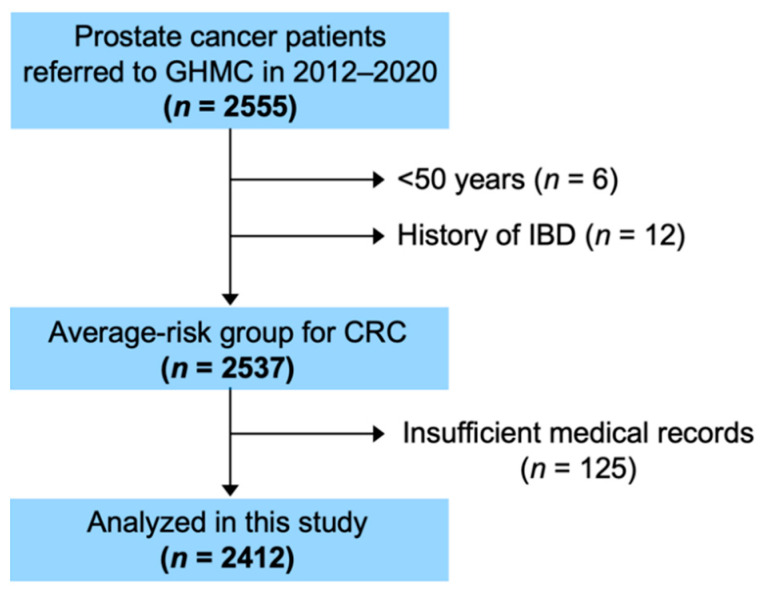
Flow diagram showing patient enrollment. GHMC, Gunma University Heavy Ion Medical Center; IBD, inflammatory bowel disease; CRC, colorectal cancer.

**Figure 2 cancers-13-04481-f002:**
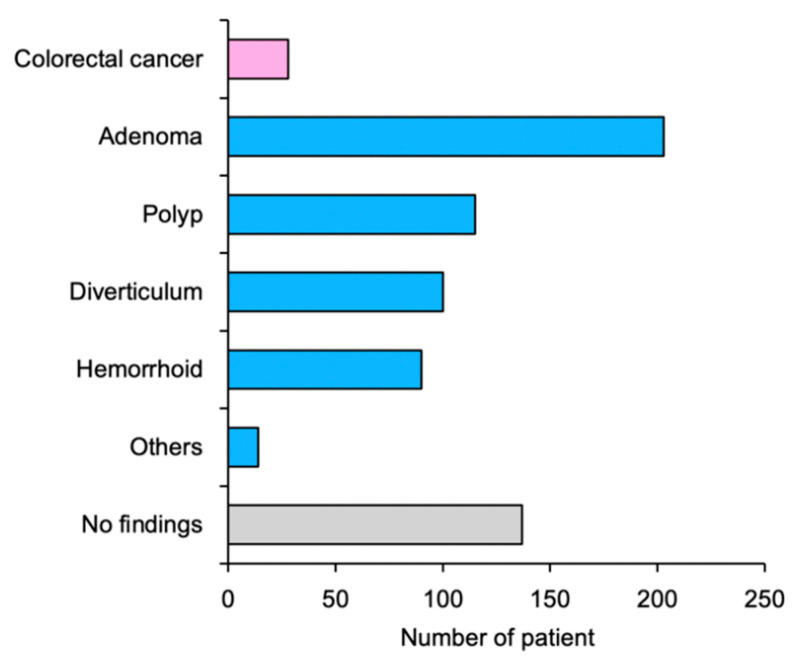
Colonoscopy findings among fecal immunochemical test-positive patients. See Section 2.2 for the definition of adenoma and polyp.

**Figure 3 cancers-13-04481-f003:**
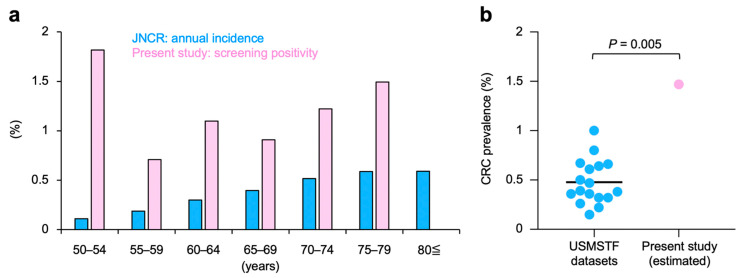
Comparison of colorectal cancer (CRC) screening outcomes between this study cohort and (**a**) the Japan National Cancer Registry (JNCR) or (**b**) a series of large-scale CRC screening studies reported in the U.S. Multi-Society Task Force (USMSTF) consensus statement. The details of the JNCR and USMSTF datasets are summarized in Section 2.1. *p*-values were assessed using the Smirnov–Grubbs test.

**Table 1 cancers-13-04481-t001:** Prevalence of CRC in average-risk adults reported in large-scale studies.

Authors	Year	Cohort	CRC			Ref.
		*n*	*n*	%	Confirmed by	
Morikawa et al.	2005	21,805	79	0.36	Colonoscopy	[11]
Imperiale et al.	2014	9899	65	0.66	Colonoscopy	[12]
Chiu et al.	2013	8822	13	0.15	Colonoscopy	[13]
Cheng et al.	2002	7411	16	0.22	Colonoscopy	[14]
Nakama et al.	1999	4611	18	0.39	Colonoscopy	[15]
Sohn et al.	2005	3794	12	0.32	Colonoscopy	[16]
Nakazato et al.	2006	3090	19	0.61	Colonoscopy	[17]
Chiang et al.	2011	2796	28	1.00	Colonoscopy	[18]
Brenner et al.	2013	2235	15	0.67	Colonoscopy	[19]
Wijkerslooth et al.	2012	1256	8	0.64	Colonoscopy	[20]
Itoh et al.	1996	27,860	89	0.32	2-year f/u	[21]
Allison et al.	1996	7493	35	0.47	2-year f/u	[22]
Launoy et al.	2005	7421	28	0.38	2-year f/u	[23]
Allison et al.	2007	5356	14	0.26	2-year f/u	[24]
Nakama et al.	1996	3365	12	0.36	2-year f/u	[25]
Parra-Blanco et al.	2010	1756	14	0.80	2-year f/u	[26]
Levi et al.	2011	1204	6	0.50	2-year f/u	[27]

CRC, colorectal cancer; Ref, reference; f/u, follow up.

**Table 2 cancers-13-04481-t002:** Association between CRC morbidity and clinical risk factors for PCa.

Prostate Cancer Risk Factors	Non-CRC Patients		CRC Patients		*p*-Value
	*n*	%	*n*	%	
Initial PSA (ng/mL)					*0.023*
<10	1510	63.4	12	42.9	
10–19.9	590	24.8	7	25.0	
20≤	282	11.8	9	32.1	
T stage					*0.20*
1	296	12.4	6	21.3	
2	1248	52.5	9	32.2	
3 + 4	835	35.1	13	46.4	
Gleason score					*>0.99*
6	191	8.0	3	14.3	
7	1352	56.8	12	57.1	
8	468	19.6	6	28.6	
9 + 10	371	15.6	7	0.0	
Tumor risk group					*>0.99*
Low	3	0.1	0	0.0	
Intermediate	1134	48.5	11	40.7	
High	1202	51.4	16	59.3	

CRC, colorectal cancer; PCa, prostate cancer; PSA, prostate-specific antigen. *p*-values, assessed by Fisher’s exact test, are shown after Bonferroni correction.

## Data Availability

The data are not publicly available due to the study protocol approved by the Institutional Ethical Review Committee of the Gunma University Hospital.

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
