# Peer review of "Colorectal Cancer Screening Outcomes of 2412 Prostate Cancer Patients Considered for Carbon Ion Radiotherapy"

_cancers, 2021, doi:10.3390/cancers13174481_

Round 1

Reviewer 1 Report

I have read with great interest this study concerning the role of colorectal cancer screening in patients candidate to receive radiotherapy for prostate cancer by means of carbon ion RT. The paper is well written and data are properly collected and presented. I believe that this article might be published in the current version with no relevant editing needed. 

Author Response

We sincerely thank the reviewer for evaluating our manuscript highly.

Reviewer 2 Report

The authors present a very interesting paper about "Colorectal Cancer Screening Outcomes of 2412 Prostate Cancer Patients Considered for Carbon Ion Radiotherapy".

The study design is absolutely appropriate and the results have been highligthed in an accurate way underlining also the potential limits.

I have only one major concern about the conclusion drawn by the authors in the abstract where they state "the prevalence of CRC in PCa patients
considered for CIRT is greater than that in general average-risk adults; therefore, pre-treatment CRC screening is beneficial for this population". 

Such conclusion is not adequately supported by the further discussion when the authors themselves admit that "It is easy to imagine that in a real-world setting a portion of cases diagnosed with CRC would have been dropped from registration in the JNCR; therefore, it would be crude to compare the 
screening outcomes of this study cohort with the incidence reported in JNCR".

I would suggest to better re-phrase the sentences without a misleading link between the indicated treatment modality (Carbon Ion Radiotherapy) and a pre-exhisting prevalence of CRC an PCa.

Author Response

Reviewer 2

The authors present a very interesting paper about "Colorectal Cancer Screening Outcomes of 2412 Prostate Cancer Patients Considered for Carbon Ion Radiotherapy". The study design is absolutely appropriate and the results have been highlighted in an accurate way underlining also the potential limits.

Response:

We sincerely thank the reviewer for evaluating our manuscript. We revised the manuscript according to the suggestion as follows.

I have only one major concern about the conclusion drawn by the authors in the abstract where they state "the prevalence of CRC in PCa patients considered for CIRT is greater than that in general average-risk adults; therefore, pre-treatment CRC screening is beneficial for this population". Such conclusion is not adequately supported by the further discussion when the authors themselves admit that "It is easy to imagine that in a real-world setting a portion of cases diagnosed with CRC would have been dropped from registration in the JNCR; therefore, it would be crude to compare the screening outcomes of this study cohort with the incidence reported in JNCR". I would suggest to better re-phrase the sentences without a misleading link between the indicated treatment modality (Carbon Ion Radiotherapy) and a pre-existing prevalence of CRC an PCa.

Response:

We agree with the reviewer's comment that the conclusion of Abstract was misleading. In accordance with the suggestion, the conclusion of Abstract was rephrased to "These data indicate the possibility that the prevalence of CRC in PCa patients is greater than that in general average-risk adults, warranting further research in a prospective setting" (lines 38–40). In addition, the conclusion of Simple Summary as well as that for main text were also revised accordingly (lines 23–25 and 244–246). We thank the reviewer for this critical comment.